# Differential Stimulation of Pluripotent Stem Cell-Derived Human Microglia Leads to Exosomal Proteomic Changes Affecting Neurons

**DOI:** 10.3390/cells10112866

**Published:** 2021-10-24

**Authors:** Anna Mallach, Johan Gobom, Charles Arber, Thomas M. Piers, John Hardy, Selina Wray, Henrik Zetterberg, Jennifer Pocock

**Affiliations:** 1Department of Neuroinflammation, UCL Queen Square Institute of Neurology, University College, London WC1N 1PJ, UK; a.mallach@ucl.ac.uk (A.M.); thomas.piers@ucl.ac.uk (T.M.P.); 2Department of Psychiatry and Neurochemistry, Institute of Neuroscience and Physiology, University of Gothenburg, S-43180 Molndal, Sweden; johan.gobom@neuro.gu.se (J.G.); h.zetterberg@ucl.ac.uk (H.Z.); 3Clinical Neurochemistry Laboratory, Sahlgrenska University Hospital, S-431 80 Molndal, Sweden; 4Department of Neurodegenerative Disease, UCL Queen Square Institute of Neurology, London WC1N 1PJ, UK; c.arber@ucl.ac.uk (C.A.); j.hardy@ucl.ac.uk (J.H.); selina.wray@ucl.ac.uk (S.W.); 5UK Dementia Research Institute at UCL, London WC1E 6BT, UK

**Keywords:** dementia, Alzheimer’s disease, exosomes, microglia, proteome, intercellular signalling

## Abstract

Microglial exosomes are an emerging communication pathway, implicated in fulfilling homeostatic microglial functions and transmitting neurodegenerative signals. Gene variants of *triggering receptor expressed on myeloid cells-2 (TREM2)* are associated with an increased risk of developing dementia. We investigated the influence of the *TREM2* Alzheimer’s disease risk variant, R47H^het^, on the microglial exosomal proteome consisting of 3019 proteins secreted from human iPS-derived microglia (iPS-Mg). Exosomal protein content changed according to how the iPS-Mg were stimulated. Thus lipopolysaccharide (LPS) induced microglial exosomes to contain more inflammatory signals, whilst stimulation with the TREM2 ligand phosphatidylserine (PS^+^) increased metabolic signals within the microglial exosomes. We tested the effect of these exosomes on neurons and found that the exosomal protein changes were functionally relevant and influenced downstream functions in both neurons and microglia. Exosomes from R47H^het^ iPS-Mg contained disease-associated microglial (DAM) signature proteins and were less able to promote the outgrowth of neuronal processes and increase mitochondrial metabolism in neurons compared with exosomes from the common *TREM2* variant iPS-Mg. Taken together, these data highlight the importance of microglial exosomes in fulfilling microglial functions. Additionally, variations in the exosomal proteome influenced by the R47H^het^ *TREM2* variant may underlie the increased risk of Alzheimer’s disease associated with this variant.

## 1. Introduction

Recent studies have identified late-onset Alzheimer’s disease (LO-AD) risk genes associated with microglia, the immune cells of the brain [1,2,3,4]. One of these genes encodes the triggering receptor expressed on myeloid cells-2 (TREM2) protein, which is expressed on microglia in the central nervous system and senses lipids, such as phospholipids, in the extracellular environment [5,6,7]. The R47H^het^ *TREM2* variant significantly increases the risk of developing LO-AD [2,8]. TREM2 deficits have been linked to reduced phagocytosis and metabolism, and an aberrant response to extracellular stimuli [9,10,11,12], impairing key microglial functions.

One key microglial function is the secretion of a range of factors, including exosomes [13,14]. Exosomes are small extracellular vesicles, which can be secreted from different cells, including microglia. Microglial exosomes have been implicated in the progression of neurodegeneration [15,16,17,18] and neuronal functioning, including neurite outgrowth [19,20]. Exosomal content has also been shown to change depending on the signals microglia receive [14].

We have recently shown that the R47H^het^ *TREM2* variant decreased the secretion of exosomes from patient-derived induced pluripotent stem cell-derived microglia (iPS-Mg) [21]. The R47H^het^ *TREM2* variant also affected the ability of exosomes to rescue neurons from cell death compared with exosomes secreted from the common variant (Cv) of *TREM2* expressing iPS-Mg [21]. 

In this study, we investigated the influence of TREM2 on exosomal content using patient iPS-Mg expressing Cv or R47H^het^ *TREM2*, by assessing the network changes of 3019 proteins rather than the analysis of the top most abundant proteins, as was determined in our previous research [21]. Thus, it was revealed that LPS treatment of iPS-Mg induced a specific inflammatory response in exosomes, whilst PS^+^ induced changes in metabolic exosomal proteins. Exosomes from R47H^het^ iPS-Mg added to neurons were less capable of supporting neuronal development and metabolic pathways compared with Cv exosomes, whilst both Cv and R47H^het^ exosomes were able to transmit inflammatory information to other homeostatic microglia in an autocrine signalling pathway.

## 2. Materials and Methods

### 2.1. Cell Lines

R47H^het^ fibroblasts from two different patients were obtained through a material transfer agreement with the University of California’s Irvine Alzheimer’s Disease Research Centre. The induced pluripotent stem cell (iPSC) lines were generated as previously described [10]. For this study, three clones of R47H^het^ were per patient line used (6 cell lines in total). In addition, the following *TREM2* common variant lines were used to generate iPS-Mg: CTRL1 (kindly provided by Prof S Wray, UCL Queen Square Institute of Neurology), CTRL2 (SBAD03, StemBANCC), CTRL3 (SFC840, StemBANCC) and CTRL4 (BIONi010-C, EBiSC). To generate iPS-neurons, the iPS line RBi001-a (Sigma Aldrich, St Louis, MO, USA) was used.

### 2.2. Cell Culture

#### 2.2.1. iPS-Mg

Using our previously published protocol and the iPSC lines, iPS-Mg were generated [10,22]. Briefly, embryoid bodies were generated from iPSC using IL-3, MCSF and β-mercaptoethanol using a previously described protocol [9,23]. Myeloid progenitors were selected and further differentiated using IL-34, MCSF and TGF-β for 2 weeks, in addition to CX3CL1 and CD200 for the final 3 days [10,22]. The generated iPS-Mg have previously been characterised in terms of their microglial signature gene expression, and display typical microglial functions, such as phagocytosis of particles, intracellular signalling and responses to inflammatory stimuli [10,24].

#### 2.2.2. iPS-Neurons

Differentiation of the iPSC into iPS-neurons was performed following a published protocol [25]. In brief, confluent iPSC were switched to N2B27 media, supplemented with 10 µM SB431542 (Tocris, Bristol, UK) and 1 µM dorsomorphin (Tocris, Bristol, UK). The N2B27 medium was composed of 1:1 DMEM-F12 and a neurobasal medium, containing 0.5× N2 supplement, 0.5× B27 supplement, 0.5× NEAA, 1 mM L-glutamine, 25 U pen/strep, 10 µM β-mercaptoethanol and 25 U insulin. After 10 days of neural induction, the cells were maintained in N2B27 without SB431542 and dorsomorphin until after Day 100, at which time the experiments were performed. The differentiation of the iPS-neurons has been previously characterised [26]. 

#### 2.2.3. SH-SY5Y Neurons

SH-SY5Y cells (a kind gift from Prof R de Silva, UCL Queen Square Institute of Neurology) were differentiated using a previously established protocol [27] by incubation with retinoic acid (RA, 10 µM) for 5 days and brain-derived neurotrophic factor (BDNF, 50 nM) for 7 days. The differentiated cells are an established neuron-like model displaying a metabolic profile similar to primary neurons and are referred to as SH-SY5Y neurons in this work [28].

### 2.3. Cell Treatment and Exosome Collection

PS^+^ cells were generated through subjecting SH-SY5Y to a heat shock for 2 h at 45 °C [9]. The heat-shocked neurons expressed phosphatidylserine on their outer plasma membrane, which we had previously verified [9,24], a known ligand of TREM2 [7,29]. Therefore, when PS^+^ neurons were added to iPS-Mg, they were termed PS^+^. 

iPS-Mg were treated with lipopolysaccharide (LPS, 100 ng/mL) or 2:1 PS^+^:iPS-Mg for 24 h before the supernatant was collected. Exosomes were extracted from the supernatant with an ExoQuick kit (System Biosciences, Palo Alto, CA, USA). The presence of exosomal markers and the size distribution of the particles were confirmed previously [21]. Extracted exosomes were lysed in a RIPA buffer and their protein content was determined by BCA analysis. 

SH-SY5Y neurons, iPS-Mg or iPS-neurons were exposed to exosomes from Cv or R47H variant iPS-Mg. For this, 6 µg of exosomal protein was added to the iPS-Mg, SH-SY5Y neurons or iPS-neurons for 24 h. For iPS-Mg experiments, exosomes from the same genetic background were added, in that Cv iPS-Mg were treated with Cv exosomes and R47H^het^ iPS-Mg were treated with R47H^het^ exosomes.

### 2.4. Proteomic Analysis

Proteomic data from 50 µg of iPS-Mg exosomes for each treatment were generated from liquid chromatography tandem mass spectrometry (LC-MS/MS) experiments described previously [21] and were further analysed to elucidate changes in the exosomal protein content depending on the cell treatment and *TREM2* variant. Different repeats were pooled into three independent samples. Functional analysis was performed using FunRich software. Heatmaps and principal component analysis (PCA) plots were produced with MATLAB. 

Based on previously published weighted gene co-expression network analyses [30,31], two signed networks were constructed, one for each of the genotypes, using different treatments as traits. The code, run through RStudio, was based on the tutorial provided with the R package. Briefly, proteins were clustered based on their dissimilarity measure before being raised to a power function, based on the assumption of a scale-free network. For both networks, this power was 30. To merge closely related modules, 30 proteins were set as a minimum module size and a threshold of 0.25 was chosen to merge closely clustered modules. Module preservation was performed using previously published papers [32], using *medianRank*, *Zsummary* and kME correlations [32,33]. 

Functional annotation of the modules was performed with the “GOenrichmentAnalysis” package, provided through RStudio, with Bonferroni correction applied to the ρ value to account for multiple comparisons. 

### 2.5. Quantitative PCR

Quantitative PCR (qPCR) experiments were performed to determine the effect of exosomes on iPS-Mg and neurons. Cells treated with exosomes as above were lysed in Trizol. RNA was extracted using the DirectZol RNA MiniPrep Plus kit (Cambridge Bioscience, Cambridge, UK, #74106). cDNA was generated with a high-capacity cDNA reverse transcription kit (Applied Biosystems, Waltham, MA, USA, #4368814), and qPCR analysis was performed using Taqman Universal Master Mix (Life Technologies, Waltham, MA, USA, #4440038) using specific primers (Table 1) in an Eppendorf Mastercycler.

### 2.6. Immunocytochemistry 

iPS-neurons were exposed to basal exosomes derived from untreated iPS-Mg carrying either the Cv or R47H^het^ *TREM2* variant, as described above, and fixed with 4% PFA with 4% sucrose for 20 min. The cells were then quenched in 50 mM NH_4_Cl for 10 min, followed by permeabilisation with 0.2% Triton X-100 for 5 min. Primary antibodies, Tuj1 (BioLegend, London, UK, #801202) and GAP43 (Abcam, Cambridge, UK, #75810), were incubated overnight at 4 °C in 5% normal goat serum. Appropriate secondary antibodies were used for 1 h at room temperature before the coverslips were mounted on slides using Vectashield with DAPI. Images were acquired on a Zeiss LSM710 confocal microscope using the LSM Pascal 5.0 software. Eight regions of interest (ROI) were taken per coverslip. For neuronal process length analysis, Tuj1 staining was skeletonised using an adapted version of a previously described protocol [34]. GAP43 levels were quantified using a Fiji macro based on a previously published study [35]. Briefly, the overlap between the GAP43 and Tuj1 images was calculated with the area of GAP43 staining within Tuj1-positive pixels normalised to the total area of Tuj1 positive pixels plotted.

### 2.7. Metabolic Assays 

Changes in metabolic activity were measured in SH-SY5Y neurons following 24 h of incubation with exosomes. Neurons were incubated with a 0.5 mg/mL MTT solution for 2 h at 37 °C. Afterwards, the MTT solution was removed and the solvent, isopropanol, was added for 15 min at room temperature to dissolve the MTT crystals. The plate was then read on a Tecan 10M plate reader. To normalise the results to the total cell number, taking potential changes in cell viability into account, 50 µL of crystal violet (Pro-Lab Diagnostics, Birkenhead, UK, #PL7000) was added to the plate, and the excess removed after 20 min. After the plate was air-dried, methanol was added and the plate was read on a Tecan 10M plate reader again. 

ATP levels were measured in SH-SY5Y neurons treated with exosomes for 24 h using a commercially available ATP kit (Thermo Fisher Scientific, Waltham, MA, USA, #A22066). The cells were lysed in a cell lysis reagent and centrifuged at 1000× *g* for 1 min, then 10 µL of the sample or ATP standard was added to 100 µL of the ATP reaction mix after the background luminescence of the ATP reaction mix was read. Luminescence was read on a Tecan 10M. Data were normalised to the protein concentration of each sample, as determined through a BCA assay. 

To ensure that the changes measured in both assays were due to changes in metabolic activity instead of cell proliferation or cell death, the crystal violet data obtained following the MTT assay were plotted (Appendix A). 

### 2.8. Endotoxin Assay

To test whether LPS was carried over in exosomal extractions from iPS-Mg exposed to LPS, endotoxin levels in the exosomes and iPS-Mg medium were measured using the HEK Blue LPS Detection Kit 2 (InvivoGen, San Diego, CA, USA) following the manufacturers’ instructions. Briefly, 25,000 HEK Blue cells were exposed to different quantities of exosomes, the medium or known levels of endotoxin. The release of secreted embryonic alkaline phosphatase from these cells following activation of NF-κB was determined through QUANTI-Blue and the plate was read on a Tecan 10M to measure absorbance.

### 2.9. Statistical Analysis

The results were represented as a mean of at least three separate experimental repeats ± SEM with a *p*-value of 0.05 or below considered statistically significant. The results were analysed using Prism Software version 5, MATLAB or R Studio. Analysis was performed on pooled control lines and pooled R47H^het^ cell lines. As described previously [21], proteomic data were log_10_-transformed prior to analysis. 

## 3. Results

### 3.1. Stimulatory Treatments Induce Specific Exosomal Proteomic Changes

Changes in exosomal proteins following treatment of microglia have been reported previously [13,14]. In our previous study, we reported the top 200 most abundant and significantly changed exosomal proteins secreted from iPS-Mg [21]; however, we did not investigate the subtle changes induced by the R47H^het^ *TREM2* variant and the effects of different iPS-Mg stimuli on the exosomal proteome. 

The overall effect of different conditions on the exosomal proteome was determined by generating a heatmap that clustered the different samples based on exosomal protein levels (Figure 1A). Therefore, samples which were similar in terms of exosomal protein levels clustered closely together, allowing for a visual inspection of overall differences among the different treatment conditions. This showed that following treatment with LPS or PS^+^, exosomes formed treatment-specific clusters, independent of *TREM2* status. Furthermore, exosomes from basal (unstimulated) iPS-Mg were different between the Cv and R47H^het^ genotypes. These differences were shown when the proposed function of the proteins was analysed (Figure 1B). Whilst exosomes from basal Cv iPS-Mg contained higher levels of proteins involved in cell growth and/or maintenance, R47H^het^ basal exosomes contained more transcription-related proteins. This is consistent with our previous finding that the most abundant proteins in exosomes from R47H^het^ iPS-Mg were involved in the negative regulation of transcription [21]. Stimulus-specific changes were associated with increases in the cytoskeletal organisation and biogenesis, and protein folding for LPS and PS^+^ respectively (Figure 1B) in both Cv- and R47H^het^-derived exosomes.

To reduce the variance in the data, principal component analysis (PCA) was applied to further analyse the clusters observed in the heatmap (Figure 1C). In line with the heatmap (Figure 1A), the samples fell into separate LPS and PS^+^ clusters (Figure 1C, blue and red, respectively). The basal Cv samples formed their own cluster (Figure 1C, grey), whilst the basal R47H^het^ samples fell into the LPS cluster. This confirmed the indication of the heatmap that the basal R47H^het^ exosomes resemble exosomes from LPS-treated iPS-Mg. In an attempt to identify further clusters, a third principal component (PC) was also plotted (Figure 1D), also showing differences between the Cv and R47H^het^ exosomes extracted from iPS-Mg treated with PS^+^ in addition to the cluster of untreated Cv exosomes and exosomes from LPS-treated iPS-Mg.

Overall, exosomal content appeared to be significantly influenced by the iPS-Mg treatment, with these two different treatments inducing different proteomic changes. Furthermore, the *TREM2* R47H^het^ variant had an effect on exosomal content from untreated iPS-Mg and iPS-Mg treated with PS^+^.

### 3.2. R47H^het^ Exosomes Contain More DAM-Associated Proteins

In addition to analysing the trends in all of the identified exosomal proteins, we investigated a subset of proteins identified in the exosomes. Due to the inverse relationship between *TREM2* expression and the development of a disease-associated microglial (DAM) signature [36], we specifically probed the exosome dataset for DAM-related proteins, based on previous publications [36,37,38]. A number of DAM-associated proteins were found in exosomes, which were plotted in a heatmap (Figure 2A).

A subset of these DAM-associated proteins (indicated in red in the dendrogram) displayed particularly high levels in basal R47H^het^ exosomes (Figure 2A). These proteins were linked to the second *TREM2*-dependent stage of DAM microglia [36]. These proteins were plotted individually (Figure 2B), confirming the previous trend shown in the heatmap that R47H^het^ exosomes contained higher levels of DAM-related proteins (Figure 2A). This suggests that basal exosomes from R47H^het^ iPS-Mg carry higher levels of DAM-like proteins than Cv iPS-Mg. To determine whether this trend was mirrored by the iPS-Mg cells themselves, the expression of DAM-related genes was assessed through qPCR (Figure 2C), which indicated that untreated R47H^het^ iPS-Mg showed higher levels of DAM genes, such as *APOE, TYROBP, AXL* and *CSF1R* [36,37,38]. This suggested that the changes seen in the exosomes are due to changes observed in the cells themselves rather than changes in the exosomal packaging.

### 3.3. Network Analysis

So far, we have analysed the levels of exosomal proteins and the similarities following different iPS-Mg treatments. To elucidate more subtle changes in the data, we used a previously published weighted gene co-expression network analysis [30,31], which has also been used for proteomic analyses [39]. This network analysis is based on the hypothesis that proteins whose levels change in a similar way may be functionally related. This allows the network to build different modules, which are made up of proteins that behave similarly and have similar functions, assigning each protein identified to a specific module. The networks were individually generated for exosomes from Cv iPS-Mg and R47H^het^ iPS-Mg (Appendix A respectively). Fourteen modules, excluding the grey module containing proteins that did not fall into any other modules, were identified in both networks. Out of the identified modules, a few modules, namely the green-yellow and salmon Cv modules and the green-yellow R47H^het^ modules, were shown to be less preserved (Appendix A), indicating that they may have been influenced by noise in the data. All of the other identified modules appear to be preserved, as indicated by the low *medianRank* and high *Zsummary* score (Appendix A), and the significant kME correlation between the two different networks (Appendix A). The correlation between the modules in both networks with the different treatments was investigated to ascertain what functions exosomes could fulfil following stimulation of the iPS-Mg (Figure 3).

In the Cv network, the modules associated with basal exosomes (brown, black and purple) were enriched for differentiation/development, signalling receptors and the activity of kinases, respectively (Figure 3A), similar to the modules which overlapped with the basal conditions from R47H^het^ exosomes, which were enriched for actin organisation and differentiation/development (Figure 3B). Following LPS treatment of iPS-Mg, exosomes from Cv iPS-Mg were highly correlated with the turquoise module, associated with the immune response (Figure 3A), whilst the modules associated with the LPS condition in the R47H^het^ network were linked with RNA binding and protein translocation (Figure 3B), perhaps indicating the different functions of exosomes from Cv and R47H^het^ iPS-Mg. Exosomes from PS^+^-treated Cv iPS-Mg were highly correlated with metabolism, RNA binding and protein translocation, similar to the modules in the R47H^het^ network that were highly correlated with the PS^+^ condition, which were linked to metabolism and proteasome activity (Figure 3B).

This analysis suggested both similarities and differences between the modules in the Cv and R47H^het^ networks; therefore, the overlap between the modules identified in the Cv and R47H^het^ networks was plotted (Figure 3C). What this shows is that some modules of the two different networks overlapped, whilst some modules were exclusive to either network, indicating differences between the two genotypes. Modules associated with the basal conditions in both networks, such as the brown Cv module and the turquoise R47H^het^ module, displayed some overlap, suggesting that some functions of the basal exosomes appear to be unaffected by the *TREM2* R47H^het^ variant at baseline, albeit with subtle differences. The turquoise immune module in the Cv network was split across two different modules in the R47H^het^ network, again suggesting subtle differences between exosomes from LPS-treated Cv and R47H^het^ iPS-Mg. The modules associated with the PS^+^ treatment displayed a strong overlap between the two different networks, indicating that there was strong link to metabolic processes in exosomes from iPS-Mg treated with PS^+^ cells, with potentially some subtle *TREM2*-specific differences.

### 3.4. Microglial Exosomes Can Transmit Inflammatory Messages

In response to LPS treatment, exosomes from iPS-Mg contained higher levels of proteins in the immune cluster identified in the network analysis (Figure 3), which is in line with previous studies showing increased inflammatory cytokines in exosomes from LPS-treated microglia-like cells [14]. As inflammatory cytokines can activate other homeostatic microglia, this was investigated further. Exosomes from LPS-treated Cv iPS-Mg significantly increased the expression of *TNF, IL6* and *IL1B* in naïve Cv iPS-Mg (Figure 4B), whilst exosomes from PS^+^-treated iPS-Mg had no effect. Exosomes from LPS-treated R47H^het^ iPS-Mg also increased *TNF*, *IL6* and *IL1B* expression in R47H^het^ iPS-Mg (Figure 4D). Interestingly, exosomes from PS^+^-treated R47H^het^ iPS-Mg also increased *TNF* levels in naïve R47H^het^ iPS-Mg (Figure 4D). Since these effects could be due to carryover of LPS or PS^+^ in the exosomes, the supernatant (SN) from the naïve iPS-Mg was spiked with 100 ng/mL LPS or 2 × 10^6^ PS^+^ cells before the exosomes were extracted. These suspensions would therefore contain any potential carryover without exosomal changes. However, naïve iPS-Mg reacted significantly more to exosomes from activated cells than to spiked exosomes (Appendix A). This was further supported by endotoxin experiments, which showed that exosomes extracted from basal and LPS-treated iPS-Mg contained negligible levels of endotoxins (Appendix A). The increase in *TNF, IL6* and *IL1B* expression in iPS-Mg exposed to LPS exosomes (Figure 4) was relative to the untreated iPS-Mg; however, the effect of basal exosomes on inflammatory cytokine expression was also considered. The addition of basal exosomes spiked with LPS or PS^+^ or untreated basal exosomes did not elicit increased levels of *TNF, IL6* or *IL1B* (Appendix A). Taken together, these data support the suggestion that exosomes from LPS-treated iPS-Mg were responsible for the increased *TNF, IL6* and *IL1B* expression in treated cells, acting as a paracrine signalling pathway to induce an inflammatory response (Figure 4).

### 3.5. Basal Cv Exosomes Can Support Neuronal Development

Because the differentiation and development module was particularly associated with baseline exosomes from Cv and R47H^het^ iPS-Mg (Figure 3A,B), and since microglia can support neuronal development [40,41], we investigated the effect of basal exosomes on the development of iPS-neurons, which closely recapitulate human neuronal development [25,42]. When iPS-neurons were exposed to basal Cv exosomes, the length of Tuj1-positive projections was increased significantly, both in comparison with non-treated neurons and those exposed to basal R47H^het^ exosomes (Figure 5B). To control for different numbers of neuronal projections identified in the different ROI, the total length of the neuronal processes was normalised to the overall Tuj1 staining (Figure 5B). In addition, we showed no significant change between both overall Tuj1 staining and voxels identified in the skeleton analysis (Appendix A), suggesting that whilst the same number of neuronal processes were imaged under the different conditions, those exposed to basal Cv exosomes formed longer projections.

One protein that is involved in neuronal process outgrowth and is found in growth cones is GAP43 [43,44]. To verify that the observed changes in neuronal processes were reflected in increased outgrowth linked to GAP43, GAP43 levels in neurons were quantified. Again, increased levels of GAP43 were observed in iPS-neurons exposed to basal Cv exosomes, compared with non-treated neurons or those treated with basal R47H^het^ exosomes (Figure 5D).

### 3.6. Metabolic Effects of Exosomes on Neurons

One of the nodes identified from the analysis of the secreted exosomes involved metabolic functions; in particular, in exosomes from iPS-Mg treated with PS^+^ cells (Figure 3). To assess whether this function identified through the network analysis is indeed translated into biological differences, exosomes were added to differentiated SH-SY5Y neurons, a neuron-like model which displays a metabolic profile similar to primary neurons [28]. Basal exosomes from Cv or R47H^het^ iPS-Mg increased the metabolic activity of SH-SY5Y neurons (Figure 6B). Interestingly, exosomes from Cv iPS-Mg treated with PS^+^ cells induced significantly higher metabolic activity in neurons than exosomes from R47H^het^ iPS-Mg (Figure 6Aiii). Furthermore, changes in ATP levels in SH-SY5Y neurons mirrored the metabolic changes (Figure 6C). Basal Cv exosomes increased ATP levels in SH-SY5Y neurons, with a similar trend observed for basal R47H^het^, albeit at lower levels (Figure 6C). When exosomes were extracted from iPS-Mg treated with either LPS or PS^+^ cells, Cv exosomes evoked significantly higher neuronal ATP levels than their R47H^het^ counterparts (Figure 6C). These findings were not due to changes in cell death or proliferation of the SH-SY5Y neurons, which was verified with propidium iodide (PI) and crystal violet (Appendix A).

To probe how exosomes might influence the mitochondrial function of neuron-like cells, the expression of different genes involved in mitochondrial biogenesis was assessed (Figure 6E). Peroxisome proliferator-activated receptor gamma coactivator 1-alpha (PGC-1α) is a key regulator of mitochondrial biogenesis, acting as a transcriptional coactivator for genes involved in mitochondrial biogenesis [45,46]. Through its co-activator, the nuclear respiratory factor 1 (NRF1), the nuclear genes required for mitochondrial biogenesis can be transcribed [47,48]. Whilst *PPARGC* showed an increased trend in expression following treatment with basal Cv exosomes (Figure 6E), *NRF1* was not affected by exosome treatment (Figure 6E). *HSPD1*, encoding for HSP60, which is involved in the folding of mitochondrial proteins and maintaining ATP production during stress [49,50,51], showed an increased trend in SH-SY5Y neurons exposed to either basal Cv or R47H^het^ exosomes (Figure 6E). Due to specific increases in both MTT metabolism and ATP levels following the addition of Cv PS^+^ exosomes (Figure 6B,C), *PPARGC, NRF1* and *HSPD1* levels following the addition of LPS or PS^+^ exosomes was also tested (Appendix A). *PPARGC* and *HSPD1* levels did not further increase in SH-SY5Y exposed to PS^+^ exosomes; rather, the levels fell.

Appendix A indicating that the increase in *PPARGC* levels (Figure 6E) was specific to basal Cv exosomes.

To verify this in another neuronal model, iPS-neurons were also exposed to basal exosomes from iPS-Mg (Figure 6F), particularly as basal exosomes had an effect on the mitochondrial metabolism in SH-SY5Y neurons. *PPARGC* expression increased in iPS-neurons exposed to both basal Cv and R47H^het^ exosomes (Figure 6F), whilst *NRF1* showed slight upregulation following treatment of iPS-neurons with basal R47H^het^ exosomes (Figure 6F), similar to *HSDP1* (Figure 6F). Contrarily, *MAPT* showed no changes following exosome treatment (Figure 6F), indicating that the overall neuronal number was not affected.

## 4. Discussion

This study investigated how the microglial exosomal proteome was influenced by the stimulus received by the microglia as well as by the *TREM2* genotype of the cells. One of the main functions of microglia is to support the differentiation and development of neurons [40,41,52], which they continue to do so throughout their lifespan. Exosomes appear to be one communication pathway through which microglia fulfil these functions, as indicated by our proteomic analysis. In contrast to our previous study, where the analysis of the proteomic dataset focussed on the top 200 most abundant proteins [21], here, we analysed subtle differences in the expression of all 3019 proteins identified by LC/MS using network analysis (Figure 3). This specifically implied that exosomes from basal iPS-Mg could influence development/differentiation, which as we further showed here, translated into a functional change in iPS-neurons exposed to basal iPS-Mg exosomes, including increased outgrowth of neuronal processes, potentially through the increased expression of GAP43 (Figure 5). As neuronal process outgrowth is linked to an increased expenditure of energy [53,54], this increase in neuronal process length was mirrored by increased mitochondrial biogenesis and ATP levels in neurons exposed to basal Cv iPS-Mg exosomes (Figure 6). Interestingly, whilst inflammatory cytokines were detected in the exosomes of activated iPS-Mg, we did not detect any neurotrophins in basal exosomes or exosomes from treated iPS-Mg, such as BDNF or nerve growth factor, which microglia have been shown to release to support neuronal functioning [55,56]. However, many other microglial factors may be involved in promoting neuronal growth; for example, through activating the MAPK and Akt signalling pathways in cerebellar granule cells, as we showed previously [40]. This is one area for future study.

### 4.1. Influence of TREM2 on Exosomes from iPS-Mg

In line with our previous study [21], basal exosomes from R47H^het^ iPS-Mg are different from basal Cv exosomes, with basal R47H^het^ exosomes resembling exosomes from LPS-treated iPS-Mg (Figure 1C,D). Others have shown that TREM2 K/O macrophages are more prone to react to low levels of TLR agonists with an exacerbated response [57,58]. Here, the results were more subtle, as one would expect from a heterozygous variant. Thus, whilst exosomes from naïve R47H^het^ are already more inflammatory (Figure 1A,C), subsequent exposure to LPS does not significantly change the exosomal profile. This could explain why there was less difference between these two groups when we previously looked at specific inflammatory markers [21].

Whether the changes in exosomal protein content were due to changes in protein packaging or underlying differences in the iPS-Mg themselves was analysed in a subset of proteins identified in this study. In line with another study [59], R47H^het^ iPS-Mg displayed an increased expression of DAM-related genes, which translated to basal R47H^het^ exosomes containing higher levels of DAM-related proteins, in comparison with basal Cv exosomes (Figure 2). Whilst other studies have found an increase in microglia expressing DAM genes in animals treated with LPS [38,60], this was not replicated in this study. This could be due to differences between murine and human microglia, or the time course of the experiments. However, a wider analysis of the proteomic content of the iPS-Mg would be needed to ascertain whether the differences in exosomal content are caused by differences in protein packaging or underlying differences in protein levels in the iPS-Mg themselves.

### 4.2. Stimulus Specificity Includes LPS-Specific Inflammatory Changes and PS^+^-Specific Metabolic Changes

Previous studies have shown that exosomal content can change depending on the stimulus the microglia receive [13,14,21,61]. Here, we compared two different stimuli for their ability to influence the exosome proteome. Treatment of iPS-Mg with LPS induced strong associations with the immune clusters consisting of inflammatory cytokines. This is in line with previous studies [14]. Furthermore, previous research has suggested that the cytokines found in exosomes can be active, independent of whether they are found in the lumen or at the exosomal surface [62]. Additionally, exosomes could be a more concentrated delivery system of cytokines to target cells [62], which could elicit a response in either neurons or other microglia exposed to these exosomes. Here, the exosomes were found to contain inflammatory cytokines, which could further activate naïve microglia (Figure 4), suggesting a possible pathway of intercellular communication between microglia.

Following treatment of iPS-Mg with PS^+^ cells, metabolic modules were altered. This could be due to the PS^+^ cells supplying iPS-Mg carrying *TREM2* variants with additional energy in the form of lipids [21,24]. This study only investigated the proteomic content of exosomes; however, previous studies have suggested that metabolites [63] and enzymes [64,65] found within the exosomes can also have an effect on cellular metabolism. Studying these aspects of exosomal cargo would further supplement the finding that the exosomal protein content can influence metabolism. These changes are also functionally relevant, as neurons exposed to exosomes from iPS-Mg treated with PS^+^ cells displayed an increase in metabolism, reflected by increased ATP levels. In particular, the increased expression of PGC-1α and *PPARGC* in neurons exposed to basal Cv exosomes suggests an increase in mitochondrial biogenesis. In addition to this, basal R47H^het^ exosomes also induced upregulation of *NRF1* and *HSPD1*, encoding for HSP60, in iPS-neurons. In the future, this could be further investigated to narrow down which aspects of mitochondrial metabolism are affected, using functional analysis of mitochondrial functions.

### 4.3. Conclusions and Outlook

Combining proteomic analysis of the exosomal content with downstream functional readouts in recipient cells has allowed us to determine important microglial exosome functions for paracrine signalling pathways, inducing the outgrowth of neuronal processes and mitochondrial biogenesis, and transmitting inflammatory signals to other microglia. R47H^het^ *TREM2* variant iPS-Mg were shown to secrete exosomes that impair these functions. These findings highlight how dementia risk genes expressed in the microglia may translate to changes in the exosomal proteome, and point to the possibility of using these as biomarkers.

## Figures and Tables

**Figure 1 cells-10-02866-f001:**
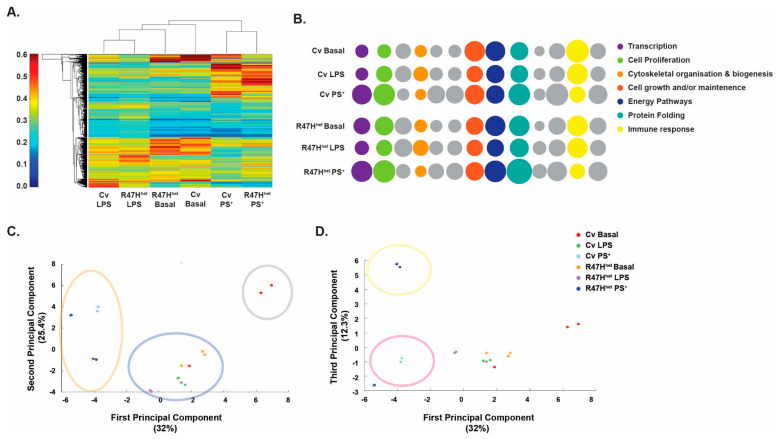
Following treatment, exosomes fell into specific clusters. Following the LC-MS/MS process, a heatmap of the exosomal proteins was constructed to analyse the effect of the different iPS-Mg treatments on exosomal proteome content (**A**). The relative abundance of proteins associated with different biological processes, identified with FunRich, was plotted for the different conditions, with the diameter of the circles representing protein abundance (**B**). Principal component analysis (PCA) was used to find clusters between the different samples (**C**,**D**). N = 3 independent samples analysed by LC-MS/MS for each condition.

**Figure 2 cells-10-02866-f002:**
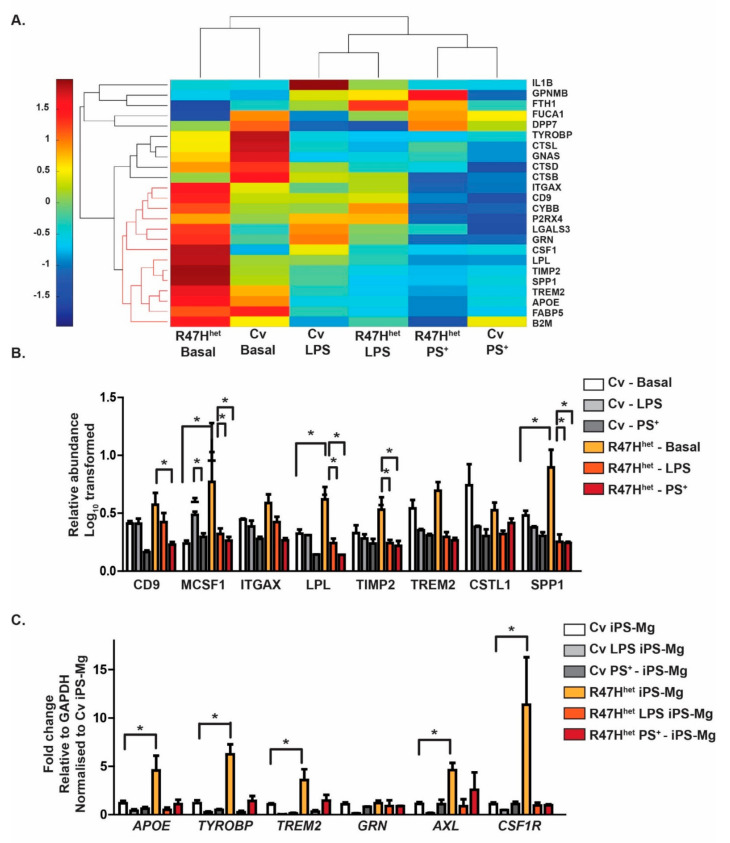
Proteins associated with the DAM microglia signature are found in exosomes. The exosomal proteomic profile was compared with a list of published DAM-related proteins [36,37,38]. R47H^het^ exosomes contained a subset of DAM-associated proteins (indicated in orange) at higher levels than Cv exosomes (**A**). These proteins were particularly enriched with the TREM2-dependent second stage of DAM [36]. The relative abundance of the DAM proteins in the exosomes was normalised to the abundance observed in Cv exosomes of these second-stage DAM-associated proteins (**B**). The expression of DAM-associated genes was measured in the iPS-Mg using qPCR to assess whether the changes observed in the exosomes reflected differences at the cellular level or in the packaging (**C**). For (**A**,**B**), N = 3 independent samples analysed through LC-MS/MS for each condition, whilst for (**C**), N = 4 independent experiments. Two-way ANOVA with * *p* < 0.05, ** *p* < 0.01.

**Figure 3 cells-10-02866-f003:**
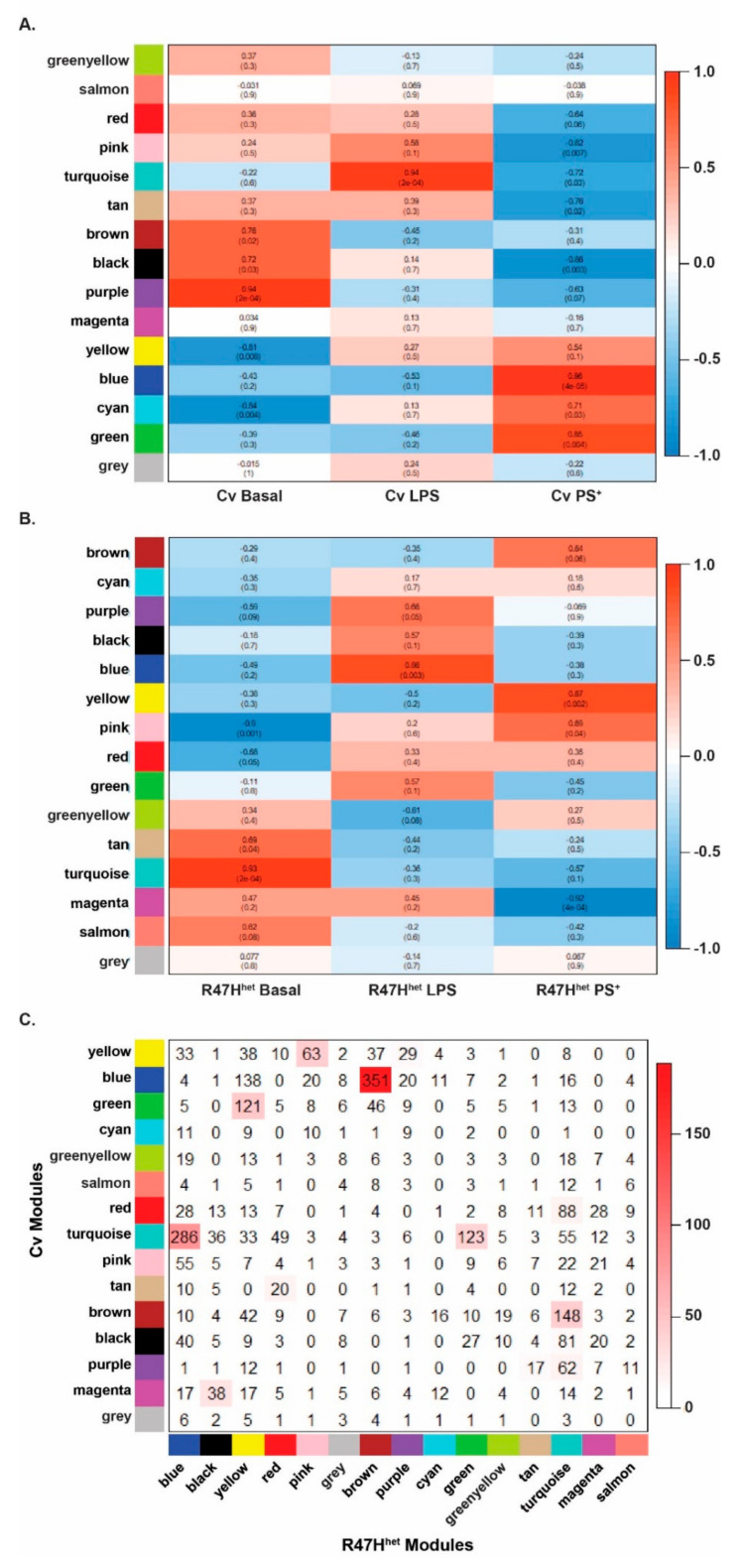
Network analysis. Based on the constructed networks, the relationship between the different iPS-Mg treatments and the identified modules was plotted. The treatment–module relationships are shown for the Cv network (**A**) and the R47Hhet network (**B**). The correlations are indicated in the boxes with the *p* values in parentheses. The direction of the correlation was colour-coded, with positive correlations indicated in red and negative correlations in blue. The overlap between the modules identified in the Cv and R47Hhet network was also plotted (**C**), with the number of shared proteins represented in the boxes and the –log (*p* value) indicated by the colouring. N = 3 independent samples analysed by LC-MS/MS for each condition.

**Figure 4 cells-10-02866-f004:**
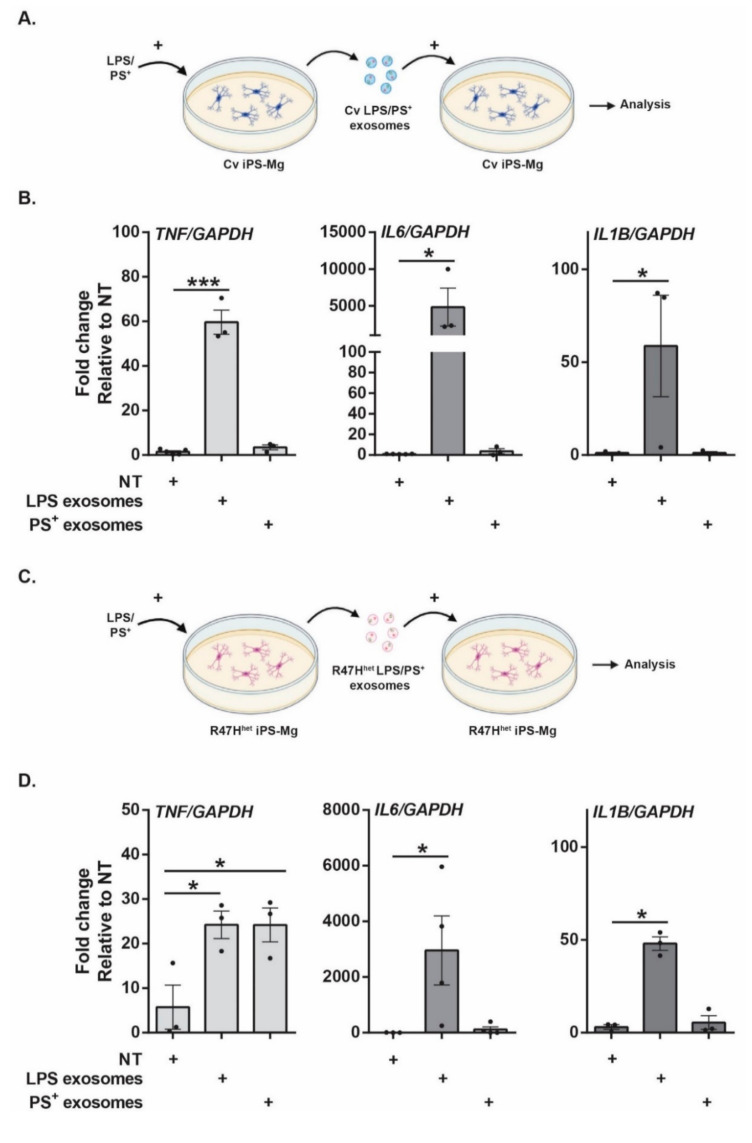
Microglial exosomes can activate naïve microglia. Based on the previous analysis (Figure 3), which showed large changes in the immune cluster following activation of iPS-Mg with LPS, the effect of iPS-Mg exosomes on naïve iPS-Mg was measured, specifically, the ability of exosomes from LPS activated iPS-Mg to transmit this information to naïve iPS-Mg. The experimental setup for (**B**,**D**) is shown in (**A**,**C**), respectively. The expression levels of *TNF, IL6* and *IL1B* were determined by qPCR and are shown for Cv iPS-Mg exposed to Cv exosomes (**B**) and for R47H^het^ iPS-Mg exposed to R47H exosomes (**D**). N = 3 separate experiments with one-way ANOVA. * *p* < 0.05, *** *p* < 0.001.

**Figure 5 cells-10-02866-f005:**
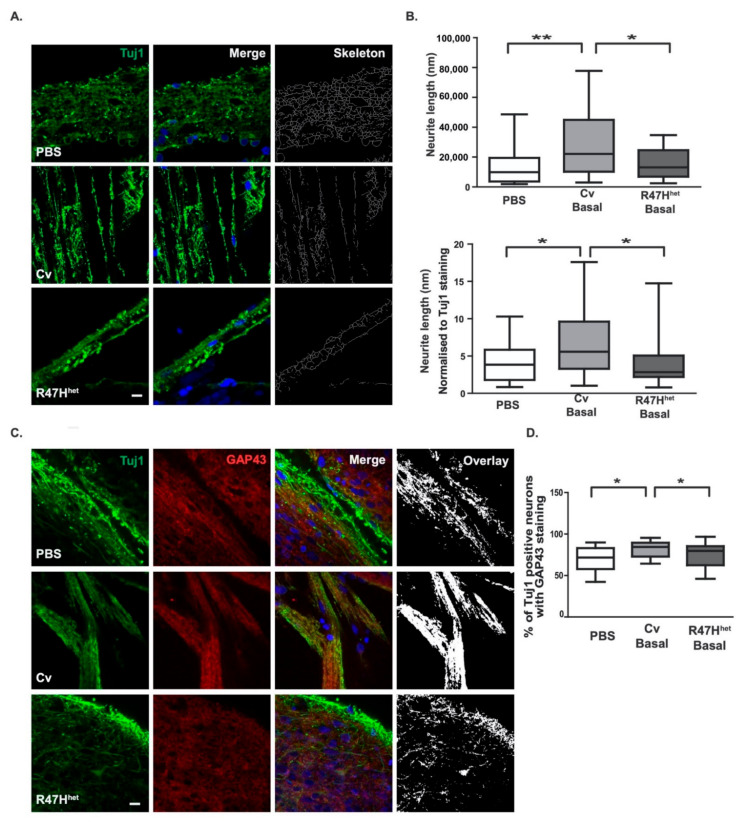
Microglial exosomes support neuronal outgrowth. The ability of exosomes to support neuronal outgrowth was analysed in iPS-neurons exposed to basal Cv and R47H^het^ exosomes. Representative images of iPS-neurons stained with Tuj1 (green) and DAPI (blue nuclear stain) (**A**). Plots of skeletonised neuronal process lengths and plots of skeletonised neuronal process lengths normalised to overall Tuj1 staining (**B**) are shown. To verify whether this change was associated with increased neuronal outgrowth, the level of GAP43 staining was quantified in Tuj1-positive projections. Representative images (**C**) following quantification of the overlap (**D**) are shown. N = 3 coverslips per condition with eight ROI analysed per coverslip with one-way ANOVA. * *p* < 0.05, ** *p* < 0.01.

**Figure 6 cells-10-02866-f006:**
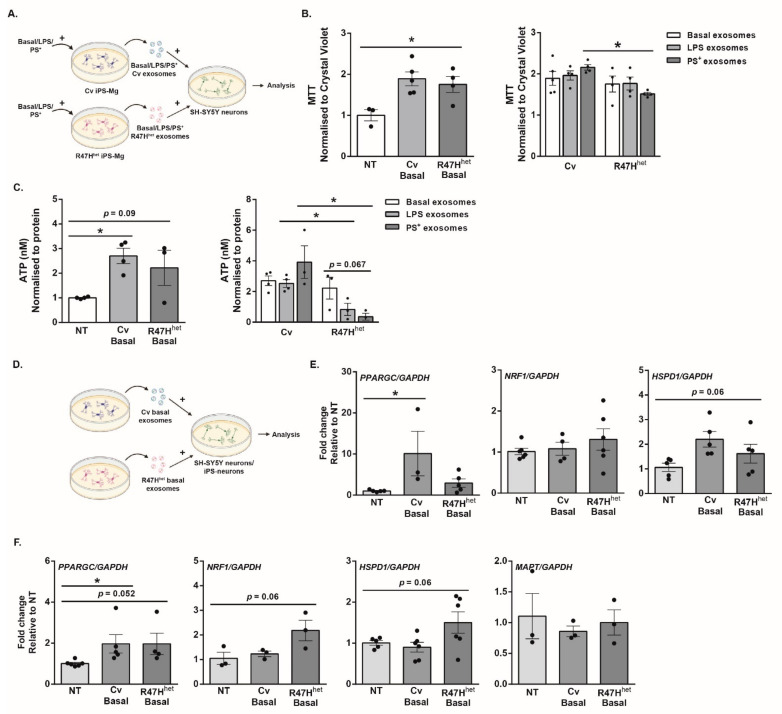
Microglial exosomes can influence neuronal metabolism. Based on the identification of the metabolism cluster and the differences between Cv and R47H^het^ exosomes in this cluster, the effect of exosomes on neuronal metabolism was tested in SH-SY5Y neurons (experimental setup: (**A**). The metabolic rate was approximated using the MTT assay (**B**). The production of ATP in the cells was measured with an ATP assay (**C**) after SH-SY5Y neurons were exposed to exosomes from Cv and R47H^het^ iPS-Mg. The relative expression of various genes in SH-SY5Y neurons and iPS-neurons after they were exposed to basal Cv and basal R47H^het^ exosomes was determined by qPCR, with the experimental setup shown in (**D**). The expression of *PPARGC, NRF1 and HSPD1* was analysed in SH-SY5Y neurons (**E**) and in iPS-neurons (**F**) in addition to the levels of *MAPT* in iPS-neurons (**F**). N = 4 independent experiments for (**B**); N = 3 independent experiments for (**C**–**F**) with one-way ANOVA (**B**,**C**,**E,F**) or two-way ANOVA (**B**,**C**). * *p* < 0.05.

**Table 1 cells-10-02866-t001:** Probes used for qPCR.

Gene Name	Identifier
*APOE*	Hs00171168_m1
*AXL*	Hs01064444_m1
*CSF1R*	Hs00911250_m1
*GAPDH*	Hs02758991_g1
*GPNMB*	Hs01095669_m1
*GRN*	Hs00963707_g1
*HSPD1*	Hs01036753_g1
*IL1B*	Hs00174097_m1
*IL6*	Hs00985639_m1
*PPARGC*	Hs01016719_m1
*TNF*	Hs01113624_g1
*TREM2*	Hs00219132_m1
*TYROBP*	Hs00182426_m1

## Data Availability

The data presented in this study are available on request from the corresponding author.

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
