# Peer review of "Differential Stimulation of Pluripotent Stem Cell-Derived Human Microglia Leads to Exosomal Proteomic Changes Affecting Neurons"

_cells, 2021, doi:10.3390/cells10112866_

Round 1

Reviewer 1 Report

The article is very interesting, with a lot of information and results.

The quality of the images and figures is very low, therefore it is difficult to understand or see the result correctly for its interpretation.

I think a figure or diagram of material and methods is necessary to be able to understand all the steps to follow and the different study groups.

The number of fibroblasts extracted is n=2, from two patients. I think it is necessary for all the work and to give it greater scientific rigor that the number of patients be increased. Is there a reason why more samples cannot be obtained?

Nor does it indicate the patient's condition with the pathology. I think it is interesting to take into account what state they are in, if they are both in advanced states, if they are similar or not.

The experiments seem well planned and executed to me, but the representation of the results in another type of more visual graph would be better to be able to see the results correctly.

Author Response

This was from a previous review. We already answered this.

Reviewer 2 Report

Below my comments on the first submission and how this has been taken care of.

Major points:

  • The quality of provided figures was very poor, including of those additional provided in a zip file. This especially for Figure 1C, Fig 2B,C, Fig 3A, B, and Fig 5.

> Still bad quality: all not sharp. Especially: Figure 1, Figure 3, Figure 5 immunostainings

  • The description of the findings is poor. For instance, the abstract is rather a listing of findings, instead of a selection of the most important findings and the conclusion that can be drawn from that. This is also the problem for the Result section, for which the results are just presented as a list of findings without a clear line of thinking.

 > Much better

  • This study uses data of a previous study from the same authors. It is not clear enough presented (introduction and discussion) how the current study builds on this, and the extent of overlap

> OK

  • Information on the quality and characterization of the generated cell lines has not been provided.

> OK

  • The common variant control lines are from another source. They therefore very well may not be good controls. Isogenic controls should have been taken along. Difference between R47H and controls may be technical.

> not addressed

  • Also, R47H fibroblast of only 2 patients have been used (of which 2x 3 lines have been created). The large variations found in the different assays may very well be reflected by the difference between these 2 patients (e.g. as possibly visible in Figure 1C).

> not addressed

  • No characterization is provided on the quality of the generated iPS-MG cells and iPSC-neurons.

> OK

  • The exosomal changes between the cell lines may also be quantitative; and underlie changes in content, this is not addressed.

> not addressed

  • Legend should only contain information to understand the presented data, not conclusions. The legends are now (badly) written as paragraphs of the result section.

> OK

  • LPS is known to induce expression of DAM genes in Microglia. This is not found here, instead DAM genes are found in the R47H lines. This is not discussed. It strongly questions the quality of the data.

> not properly addressed

  • Why are iPS-neurons used to study neuronal development, and SH-SY5Y cells to study neuronal metabolism? For both the iPSC-neurons seem a better system reflecting neuronal biology.

> not addressed

  • Fig5 is stated to depict neuronal outgrowth, but no neurites are visible. MAP2 is better used to stain and quantify length of dendrites.

> not addressed: terrible quality pictures: no neurites visible.

  • The found differences in Fig 6 between CV_PS+ and R47H_PS+ are not convincing. There is no difference between CV_basal and CV-PS+, nor between R47H_basal and R47H_PS+. The only remarkable thing is that variation is very small for the PS+ samples (which may make it sign), while variation is large for the other samples. It is also not clear what the points (3 to 6) represent; there are 2 patient lines with 3 lines each.

> not addressed. Also, It is stated: N=4 independent experiments for A. Which means 4 different culturing experiments (done at different moments), this is very unlikely to give this small variation as found for instance in PS+ samples?

New comments:

16) The legend of Figure 2 is now placed for a second time in the middle of the Discussion.

17) >‘LPS’ missing in first sentence Results.

Author Response

(The authors gave the same response as above.)

Reviewer 3 Report

There are no significant improvement in the revised manuscript. The author neglect the reviewer's adivce and made no change.

Author Response

(The authors gave the same response as above.)

Reviewer 4 Report

This manuscript maybe seems interesting, but it presents a serious of flaws which undermines its quality:

1)    Clear redundant repetition of protocol Page 2-3:

SH-SY5Y cells (a kind gift from Prof R de Silva, UCL Queen Square Institute of Neurology), were differentiated using a previously established protocol [28] by incubation with retinoic acid (RA, 10 µM) for 5 days and brain derived neurotrophic factor (BDNF, 50 nM) for 7 days. These cells were differentiated using a previously established protocol [28] by incubation with retinoic acid (RA, 10 µM) for 5 days and BDNF (50 nM) for 7 days. The differentiated cells are an established neuron-like model displaying a metabolic profile similar to primary neurons and referred to as SH-SY5Y neurons in this paper [29].

2)    Page 3: Clear grammatical error
 Proteomic data from 50 µg of iPS-Mg exosomes for each treatment was generated from liquid chromatography tandem mass spectrometry (LC-MS/MS) experiments described previously [22] were further analysed to elucidate changes in exosomal protein content depending on cell treatment and TREM2 variant. 

3)    Figures in the paper present a very bad resolution which makes it difficult to read or look at (i.e., Figure1. 3, 4, 5, 6).
It is like when you get dressed up and go out: if you have a big stain or rip on your clothes people will look at you differently and not consider you as smart as you want to appear. Same here, if you present figures with a resolution this bad you can’t expect the reviewers to see if or how outstanding is your data. I would suggest to use a PNG format from the beginning if the size of the images are too high. It seems a jpeg which every time is opened it loses resolution
Supplementary Figures on the other hand are very good which does not explain why the figures presented are so bad.

4)    In Figure 1,4, 5, 6, in particular, why would the authors label panels of the figures ai aii aiii bi etc?  I guess to say that they were connected, but it would have been easier to uniform to a, b, c,d,etc…
5)    Fibroblast of only 2 patients have been used. Moreover, the authors use a very low N for independent experiments. I may understand why the authors choose to use only two patients (although I do not agree) but the cell cultivated can be increased in number. The variability is too low. It should definitely be increased.
6)    Page 16 in the discussion: It is not clear “Exosomes appear to be one pathway through which microglia fulfil these functions, as indicated from our proteomic analysis.” Which pathway?

7)    On Page 16 the caption of Figure 2 seems just placed there. Why?

Author Response

Comments and Suggestions for Authors

This manuscript maybe seems interesting, but it presents a serious of flaws which undermines its quality:

We thank the reviewer for noting the novelty of this paper.

1)    Clear redundant repetition of protocol Page 2-3:

SH-SY5Y cells (a kind gift from Prof R de Silva, UCL Queen Square Institute of Neurology), were differentiated using a previously established protocol [28] by incubation with retinoic acid (RA, 10 µM) for 5 days and brain derived neurotrophic factor (BDNF, 50 nM) for 7 days. These cells were differentiated using a previously established protocol [28] by incubation with retinoic acid (RA, 10 µM) for 5 days and BDNF (50 nM) for 7 days. The differentiated cells are an established neuron-like model displaying a metabolic profile similar to primary neurons and referred to as SH-SY5Y neurons in this paper [29].

The redundancy has been removed.

2)    Page 3: Clear grammatical error
 Proteomic data from 50 µg of iPS-Mg exosomes for each treatment was generated from liquid chromatography tandem mass spectrometry (LC-MS/MS) experiments described previously [22] were further analysed to elucidate changes in exosomal protein content depending on cell treatment and TREM2 variant. 

This sentence has now been changed to correct this grammatical error.

3)    Figures in the paper present a very bad resolution which makes it difficult to read or look at (i.e., Figure1. 3, 4, 5, 6).
It is like when you get dressed up and go out: if you have a big stain or rip on your clothes people will look at you differently and not consider you as smart as you want to appear. Same here, if you present figures with a resolution this bad you can’t expect the reviewers to see if or how outstanding is your data. I would suggest to use a PNG format from the beginning if the size of the images are too high. It seems a jpeg which every time is opened it loses resolution. Supplementary Figures on the other hand are very good which does not explain why the figures presented are so bad.

We think the reviewer may not have seen the zip folder containing the individual figures. The figures lose resolution when embedded into the text, which we pointed out in the first review and to the Editor, however we did save them separately in a zip folder with the 300dpi resolution required by the journal. As the reviewer points out, the supplementary figures are better, this is because they are not embedded in the doc.

4)    In Figure 1,4, 5, 6, in particular, why would the authors label panels of the figures ai aii aiii bi etc?  I guess to say that they were connected, but it would have been easier to uniform to a, b, c,d,etc…

As the reviewer noticed, the format was chosen to indicate connected sub-figures. We believe that this makes it easier indeed to reference the individual sub-figures in both the text and the figure legends.

5)    Fibroblast of only 2 patients have been used. Moreover, the authors use a very low N for independent experiments. I may understand why the authors choose to use only two patients (although I do not agree) but the cell cultivated can be increased in number. The variability is too low. It should definitely be increased.

Because the R47Hhet variant is a rare variant in the population and the aim of the study was to specifically study patient-derived cell lines (as opposed to isogenic lines), only 2 patient lines were available. However, to account for potential variability in generating the iPSC, three clones were used per patient line, as specified on page 3 of the method section.

6)    Page 16 in the discussion: It is not clear “Exosomes appear to be one pathway through which microglia fulfil these functions, as indicated from our proteomic analysis.” Which pathway?

We apologise for the confusion. The manuscript has been amended to specify that we meant that exosomes are one communication pathway through which microglia can influence neuronal functions.

7)    On Page 16 the caption of Figure 2 seems just placed there. Why?

We apologise for this. We intended for Figure 2 to be placed just below the caption, however due to editing errors, the figure moved unfortunately. This has now been addressed and the figure placed below the caption.

Reviewer 5 Report

The manuscript ""Differential stimulation of pluripotent stem cell-derived human microglia leads to exosomal proteomic changes affecting neurons"  by Mallach et al. is an interesting and elegant study highlighting "the importance of microglia exosomes in fulfilling microglia function" and as well how variations in the exosomal proteome can potentially increase the risk of Alzheimer. The MS is well written and with different interesting techniques.

The only concern is that authors did not discuss  the limitations of those techniques and it would be worthy if they can include some paragraphs explaining those limitations and how to check the effectiveness and strength as well as the adequacy for demonstrating each hypothesis. 

In the last two lines of the first paragraph of the discussion section could shortly explain or describe the main microglial factors which promote neuronal growth (which they published in 2004?).

In addition, the quality of the figure is medium and it should be improved.

On page 5, in the first line of the first paragraph of the Results section, they should remove "with" (microglia with have).

Author Response

The manuscript ""Differential stimulation of pluripotent stem cell-derived human microglia leads to exosomal proteomic changes affecting neurons"  by Mallach et al. is an interesting and elegant study highlighting "the importance of microglia exosomes in fulfilling microglia function" and as well how variations in the exosomal proteome can potentially increase the risk of Alzheimer. The MS is well written and with different interesting techniques.

We thank the reviewer for this comment.

The only concern is that authors did not discuss  the limitations of those techniques and it would be worthy if they can include some paragraphs explaining those limitations and how to check the effectiveness and strength as well as the adequacy for demonstrating each hypothesis. 

A discussion of limitations included on Pages 20, 21. Thus “ Whilst other studies have found an increase in microglia expressing DAM genes in animals treated with LPS [39,61] this was not replicated in this study. This could be due to differences between murine and human microglia, or the time course of the experiments. However, a wider analysis of the proteomic content of the iPS-Mg would be needed to ascertain whether the differences in exosomal content are caused by differences in protein packaging or underlying differences in protein levels in the iPS-Mg themselves.”

In the last two lines of the first paragraph of the discussion section could shortly explain or describe the main microglial factors which promote neuronal growth (which they published in 2004?).

In our previous study we identified pathways activated and more information was added to make this more clear. Thus “However, many other microglial factors may be involved in promoting neuronal growth, for example through activating the MAPK and Akt signalling pathways in cerebellar granule cells as we showed previously [41], and this is one area of future study”. 

In addition, the quality of the figure is medium and it should be improved.

In addition to the embedded figures, we attached a zip folder containing the individual figures saved at 300dpi as required by the journal. Perhaps the reviewer missed this? We highlighted this problem to the editor and it was also pointed out by another reviewer that the supplementary figures are good quality. This is because they are not embedded in the main doc. We do not understand this but believe it is not our fault. We would prefer to upload the figures independently.

On page 5, in the first line of the first paragraph of the Results section, they should remove "with" (microglia with have).

This has now been removed.

Round 2

Reviewer 4 Report

The authors have improved the paper although the quality of figures remains bad. I read the comment where the authors state that they uploaded good quality images in a zip file asking if the reviewer missed it. To be sincere we first see the paper you upload. If you go around dressed with ragged closed we will only see what you show us even if you say you have the good clothes in your bag. I would recommend the authors to improve the quality of images when they upload the figure with the manuscript. It is really not that difficult, only a bit of more your time and less waste of ours.

This manuscript is a resubmission of an earlier submission. The following is a list of the peer review reports and author responses from that submission.

Round 1

Reviewer 1 Report

1. This manuscript is a well prepared text but still need a complete proofreading in English grammar and some spelling. 
2. Some statement is not appropriate scientific language, eg. 'Figure 1.–Following treatments, exosomes fall into specific clusters.' The author should use plain scientific accent to illustrate the data.
3. 'Signalling' is not a typical word in biological 'signaling' research. The author should use the appropriate scientific spelling.
4. The author should not include too much methods and results informations in the figure legends. 
5. A major defect in this research is the design of 'Microglial exosomes can transmit inflammatory messages'. LPS can be transfered in exosomes purification and treatment analysis. The author can use LPS antibodies or TLR4 antagonist to block it. This section should be supplied more controls to draw conclusion. Another choice is that the author can delete this section or do not draw conclusion based on current simple data.
6. Trem2 mediates anti-inflammatory response in microglia during neurodegenerations. However, the in vitro treatment using 100 ng/ml LPS induce an ultrastrong pro-inflammatory activation. This definitely can not mimic the in vivo neuroinflammation in neurodegeneration. This may also explain why the Trem2 in this research design can not show anti-inflammatory effect in in vitro microglia. The author should include more discussion and references on this design limitation.

Reviewer 2 Report

It is a really interesting and innovative research. 
It also has numerous techniques used to find the results.
The introduction should be explained in more detail in order to understand it better.
In general the graphs and figures should be improved. Especially figure 1, figure 2 and figure 3. The immunohistochemistry images are not clear either. In general the figures could be re-staged better and could be obtained with better clarity or resolution. 
It is difficult to read the text of the results and therefore the conclusions are difficult to understand.

Reviewer 3 Report

This manuscript is in principle interesting because it may provide readers an overview of proteomic changes that occur in microglial exosomes from TREM2 variants, and/or after stimulation by LPS or apoptotic neuros (PS+ membranes).

There are however a number of major points:

  • The quality of provided figures was very poor, including of those additional provided in a zip file. This especially for Figure 1C, Fig 2B,C, Fig 3A, B, and Fig 5.
  • The description of the findings is poor. For instance, the abstract is rather a listing of findings, instead of a selection of the most important findings and the conclusion that can be drawn from that. This is also the problem for the Result section, for which the results are just presented as a list of findings without a clear line of thinking.
  • This study uses data of a previous study from the same authors. It is not clear enough presented (introduction and discussion) how the current study builds on this, and the extent of overlap.
  • Information on the quality and characterization of the generated cell lines has not been provided.
  • The common variant control lines are from another source. They therefore very well may not be good controls. Isogenic controls should have been taken along. Difference between R47H and controls may be technical.
  • Also, R47H fibroblast of only 2 patients have been used (of which 2x 3 lines have been created). The large variations found in the different assays may very well be reflected by the difference between these 2 patients (e.g. as possibly visible in Figure 1C).
  • No characterization is provided on the quality of the generated iPS-MG cells and iPSC-neurons.
  • The exosomal changes between the cell lines may also be quantitative; and underlie changes in content, this is not addressed.
  • Legend should only contain information to understand the presented data, not conclusions. The legends are now (badly) written as paragraphs of the result section.
  • LPS is known to induce expression of DAM genes in Microglia. This is not found here, instead DAM genes are found in the R47H lines. This is not discussed. It strongly questions the quality of the data.
  • Why are iPS-neurons used to study neuronal development, and SH-SY5Y cells to study neuronal metabolism? For both the iPSC-neurons seem a better system reflecting neuronal biology.
  • Fig5 is stated to depict neuronal outgrowth, but no neurites are visible. MAP2 is better used to stain and quantify length of dendrites.
  • The found differences in Fig 6 between CV_PS+ and R47H_PS+ are not convincing. There is no difference between CV_basal and CV-PS+, nor between R47H_basal and R47H_PS+. The only remarkable thing is that variation is very small for the PS+ samples (which may make it sign), while variation is large for the other samples. It is also not clear what the points (3 to 6) represent; there are 2 patient lines with 3 lines each.